# Exercise Effects on Bone Mineral Density in Men

**DOI:** 10.3390/nu13124244

**Published:** 2021-11-26

**Authors:** Michelle Mages, Mahdieh Shojaa, Matthias Kohl, Simon von Stengel, Clemens Becker, Markus Gosch, Franz Jakob, Katharina Kerschan-Schindl, Bernd Kladny, Nicole Klöckner, Uwe Lange, Stefan Middeldorf, Stefan Peters, Daniel Schoene, Cornel C. Sieber, Reina Tholen, Friederike E. Thomasius, Michael Uder, Wolfgang Kemmler

**Affiliations:** 1Institute of Medical Physics, Friedrich-Alexander University Erlangen-Nürnberg, 91052 Erlangen, Germany; magesmichelle@gmail.com (M.M.); mahdieh.shojaa@imp.uni-erlangen.de (M.S.); simon.von.stengel@imp.uni-erlangen.de (S.v.S.); daniel.schoene@fau.de (D.S.); 2Department Population-Based Medicine, Institute of Health Science, University Hospital Tübingen, 72076 Tübingen, Germany; 3Department of Medical and Life Sciences, University of Furtwangen, 78056 Villingen-Schwenningen, Germany; Matthias.Kohl@hs-furtwangen.de; 4Robert-Bosch-Krankenhaus, Geriatrie und Geriatrische Rehabilitation, 70376 Stuttgart, Germany; clemens.becker@rbk.de; 5Klinikum Nurnberg, Paracelsus Medizinische Privatuniversität, 90419 Nürnberg, Germany; Markus.Gosch@klinikum-nuernberg.de; 6Bernhard Heine Zentrum für Bewegungsforschung, University of Würzburg, 97074 Würzburg, Germany; f-jakob.klh@uni-wuerzburg.de; 7Austrian Society for Bone and Mineral Research (ÖGKM), A-1090 Wien, Austria; katharina.kerschan-schindl@meduniwien.ac.at; 8German Society for Orthopaedics and Trauma (DGOU), 10623 Berlin, Germany; Bernd.Kladny@fachklinik-herzogenaurach.de; 9Deutsche Rheuma-Liga Bundesverband e.V., 53111 Bonn, Germany; nicole.kloeckner@t-online.de; 10German Society for Physical and Rehabilitative Medicine (DGMPR), 01067 Dresden, Germany; U.Lange@kerckhoff-klinik.de; 11International Musculoskeletal Pain Society (IGOST), 88212 Ravensburg, Germany; SMiddeldorf@schoen-klinik.de; 12Deutscher Verband für Gesundheitssport und Sporttherapie e. V. (DVGS), 50354 Hürth, Germany; stefan.peters@dvgs.de; 13European Geriatric Medicine Society (EuGMS), Institute for Biomedicine of Aging, FAU Erlangen-Nürnberg, 90419 Nürnberg, Germany; cornel.sieber@fau.de; 14Deutscher Verband für Physiotherapie (ZVK) e.V., 50679 Cologne, Germany; reinatholen@nord-com.net; 15Osteology Umbrella Association Germany (DVO), 67659 Kaiserslautern, Germany; FE_Thomasius@web.de; 16Institute of Radiology, FAU-Erlangen-Nürnberg and University Hospital Erlangen, 91054 Erlangen, Germany; michael.uder@uk-erlangen.de

**Keywords:** bone mineral density, exercise, men, overview

## Abstract

In contrast to postmenopausal women, evidence for a favorable effect of exercise on Bone Mineral Density (BMD) is still limited for men. This might be due to the paucity of studies, but also to the great variety of participants and study characteristics that may dilute study results. The aim of the present systematic review and meta-analysis was to evaluate the effect of exercise on BMD changes with rational eligibility criteria. A comprehensive search of six electronic databases up to 15 March 2021 was conducted. Briefly, controlled trials ≥6 months that determined changes in areal BMD in men >18 years old, with no apparent diseases or pharmacological therapy that relevantly affect bone metabolism, were included. BMD changes (standardized mean differences: SMD) of the lumbar spine (LS) and femoral neck (FN) were considered as outcomes. Twelve studies with 16 exercise and 12 control groups were identified. The pooled estimate of random-effect analysis was SMD = 0.38, 95%-CI: 0.14–0.61 and SMD = 0.25, 95%-CI: 0.00–0.49, for LS and FN, respectively. Heterogeneity between the trials was low–moderate. Funnel plots and rank and regression correlation tests indicate evidence for small study publication bias for LS but not FN-BMD. Subgroup analyses that focus on study length, type of exercise and methodologic quality revealed no significant difference between each of the three categories. In summary, we provided further evidence for a low but significant effect of exercise on BMD in men. However, we are currently unable to give even rough exercise recommendations for male cohorts.

## 1. Introduction

Many guidelines on osteoporosis and fracture prevention consider physical exercise as the most effective non-pharmacologic agent for increasing bone strength and reducing falls (e.g., [1,2,3])**.** However, in contrast to female cohorts [4,5], evidence for a favorable effect of exercise on bone mineral density (BMD) in men is still limited (review in [6,7,8,9]). Recently, two systematic reviews and meta-analyses of randomized controlled trials reported data on physical activity/exercise effects on BMD in men 18 years and older [6,8]. Ashe et al. [6] reported significant exercise effects on BMD at the total hip but little or no effect of exercise on the adjacent femoral neck (FN)-BMD. In contrast, the meta-analysis of Hamilton et al. [8] indicates significant evidence for an exercise effect on FN-BMD. Of importance, both studies do not observe any relevant beneficial effect of exercise on BMD at the lumbar spine (LS). Considering this brand-new meta-analytic data, one justifiably wonders about the rationale for another meta-analysis in the area of exercise, BMD and men. However, trivially, the results of meta-analyses predominately depend on the studies included. Apart from new study results, our eligibility criteria substantially differ from both studies. This particularly relates to pre-study exercise status, prevalent diseases/conditions and pharmacologic therapy with potential impact on exercise effects (i.e., differences in BMD-changes between exercise and control) on BMD. However, applying too critical a set of criteria aggravates the paucity of exercise studies with men performed in the gynocentric field of osteoporosis. Thus, we aimed to apply a balanced compromise between eligibility and number of studies included. This embraces the approach to exclude studies that focus on cohorts with diseases (e.g., inflammatory diseases) and pharmaceutic therapy (e.g., androgen deprivation therapy) with proven negative impact on exercise effects on bone. In summary then, the aim of the present systematic review and meta-analysis was to evaluate the effect of exercise on BMD at the lumbar spine and proximal femur ROIs in men applying reasonable eligibility criteria. Additionally, we intended to identify study and exercise characteristics with impact on BMD.

## 2. Materials and Methods

### 2.1. Literature Search

The current systematic review and meta-analysis follows the Preferred Reporting Items for Systematic Reviews and Meta-Analyses (PRISMA) Statement [10] and was registered in the international prospective register of systematic reviews. (PROSPERO; ID: CRD42021233194).

Although our 2018 systematic review on exercise effects in men [9] identified eligible articles before 2018, an new extensive search of electronic databases was performed through PubMed, Science direct, Scopus, Web of Science, Cochrane and ERIC for all articles published in German or English language up to March 15, 2021. A standard search protocol was developed and controlled vocabulary (Mesh term for MEDLINE) was applied. In order to include all the relevant studies, the following key words and their synonyms were used: “Exercise” or “Physical activity” or “Exercise training” or “Resistance training” or “Training”) AND (“men” or “male”) AND (“Bone” or “Bone mass” or “Bone density” or “Bone mineral content” or “Bone mineral density” or “BMD”). Additionally, the reference lists of the identified studies were reviewed to identify further relevant articles. Duplicate publications were identified by comparing author names, type of intervention, intervention duration and date of publication. If further information was needed, the authors were contacted via e-mail. In summary, four authors were contacted, three of them finally responded. One author submitted data for their male cohort [11], two authors were unable to provide further study information.

### 2.2. Inclusion and Exclusion Criteria

We included studies/study arms with (1) randomized and non-randomized controlled trials with at least one exercise group as an intervention versus one control group with sedentary/habitual active lifestyle without exercise or with sham exercise. (2) ≥6 months intervention duration, (3) Areal BMD of the LS, femoral neck (FN) and/or total hip (tH) region at baseline and study end as determined by (4) dual-energy X-ray absorptiometry (DXA) or dual-photon absorptiometry (DPA). We excluded studies with (1) diseases/conditions with relevant impact on bone metabolism (e.g., inflammatory diseases), (2) pharmaceutic therapy that relevantly affects bone metabolism (e.g., androgen deprivation therapy), (3) mixed-gender cohorts without separate BMD analysis for men, (4) double/multiple publications from one study and preliminary data from subsequently published trials; (5) Review articles, case reports, editorials, conference abstracts, and letters were not considered. (6) We also excluded studies with participants with pre-training exercise habits close to the volume and intensity applied by the intervention protocol.

### 2.3. Data Extraction

Titles and abstracts were screened by an independent reviewer (MM) carefully supervised by a second rater (WK). Full-text articles of relevant studies were evaluated by two reviewers (MM and WK) independently and they extracted data from included studies. In the case of disagreement, the third reviewer was consulted until a consensus was reached. An extraction form was utilized to consider the relevant data including publication information (i.e., name of the first author and year of publication), study details (i.e., study duration, initial sample size of the participants, dropout rate), participants’ characteristics (i.e., age, BMI, health-, BMD- and exercise status, medication with impact on bone, nutrition, dietary supplementation) (Table 1), methodologic quality aspects (Table 2), exercise characteristics (i.e., type of exercise, progression of intensity, frequency, setting and supervision, duration, sets and repetition, site specificity), and adherence to exercise (including number of withdrawals) (Table 3).

### 2.4. Outcome Measures

(Areal) BMD of the LS and/or the proximal femur regions “total hip” and/or the FN, as determined by dual-energy X-ray absorptiometry (DXA), are used as outcome measures. BMD assessment must be reported at least for one of the regions listed above at baseline and follow-up assessment at the end of intervention period. 

### 2.5. Quality Assessment

Two reviewers (MM, WK) assessed the included articles independently for risk of bias using the Physiotherapy Evidence Database (PEDro) scale risk of bias tool [23]. Additionally, both reviewers used the “Tool for the Assessment of Study Quality and reporting in Exercise” (TESTEX) [24] to determine study quality and reporting. Disagreements between the two reviewers were resolved by consensus in consultation with a third independent reviewer (SvS).

Studies were screened for potential selection bias, performance bias, detection bias, attrition bias and reporting bias using the 11 criteria of the PEDro-scale (Table 2). In total, the scale scores 10 items. Using the TESTEX criteria [24] as well allowed us to consider the following additional five exercise-relevant aspects: adverse effects, attendance report, activity monitoring in control groups, progression of relative exercise intensity and exercise volume (Table 2).

### 2.6. Data Synthesis

For sub-analyses, the intervention duration was stratified as 6–9 months, 10–16 months and >16 months. We also categorized the included studies by their type of exercise into three sub-groups: (a) resistance training (RT), (b) weight bearing exercises (WB), (c) RT + WB. Finally, we classified the studies according to the PEDro-Score (low <5 vs. moderate 5–6 vs. high ≥7 score points) [25]. If the studies presented a confidence interval (CI) or standard errors (SE), these were converted to standard deviation (SD) with standardized formulas [26].

### 2.7. Statistical Analysis

Briefly, a random-effects meta-analysis was conducted by using the metafor package [27] included in the statistical software R [28]. Effect size (ES) values were presented as standardized mean differences (SMDs) along with the 95% confidence interval (95%-CI). In addition to the traditional random-effects model, we applied the more robust inverse heterogeneity (IVhet) model proposed by Doi et al. [29]. A priori sensitivity analysis was applied to determine whether the overall result of the analysis is robust to the use of the imputed correlation coefficient (minimum, mean or maximum). Heterogeneity between the studies was checked using the Cochran Q test and I^2^ statistics (0–40%: low, 30–60%: moderate, 50–90%: substantial heterogeneity [26]). For those studies with two exercise groups (i.e., [15,16,20,22]), the control group was split into two smaller groups for comparison against each intervention group [26]. Funnel plots with regression test and the rank correlation between effect estimates and their standard errors using the t-test and Kendall’s τ statistic, respectively, were applied to explore potential small study/publication bias. To adjust the results for possible publication bias, we also conducted a trim and fill analysis using the L0 estimator proposed by Duval et al. [30]. In parallel, we used Doi plots and the Luis Furuya-Kanamori index (LFK index) [31] to check asymmetry. Finally, we applied influence analyses excluding two “critical” studies [11,14] from the analysis for BMD- LS and FN. A *p*-value < 0.05 was considered as the significance level for all tests. Subgroup analyses were applied for intervention length and type of exercise. 

## 3. Results

### 3.1. Study Characteristics and Quality Assessment

In total, our search identified twelve studies [11,12,13,14,15,16,17,18,19,20,21,22] (Figure 1) with 16 exercise and 12 control groups. The pooled number of participants was 823 (intervention groups: 461, control groups: 362). Table 1 presents the study and participant characteristics of the eligible studies that determined the effect of exercise on bone mineral density among men. In addition to the information given in Table 1, all but one study [22] focused on Caucasian men 36 ± 7 [14] to 79 ± 5 years [18] old. Although not always consistently stated, BMD was addressed as the primary outcome in 11 of 12 studies [11,12,13,14,15,16,17,18,19,21,22]. 

The methodological quality of the studies was rated using the PEDro and the TESTEX scale, both particularly dedicated to exercise studies. Score points vary between three and eight from a maximum of 10 points for PEDro and seven–fourteen from a maximum of 15 for the TESTEX score (Table 3). However, successful blinding of trainers and/or participants is almost impossible in conventional exercise studies; thus, maximum score-points were rather eight for PEDro and 14 for TESTEX. 

### 3.2. Intervention Characteristics

#### 3.2.1. Protein, Vitamin-D and Calcium Supplementation; Nutrition

Only one study [18] provided all participants with a maximum of 1000 mg/d Ca and 1400 IU/d Vit-D according to Ca-questionnaires and serum 25 OHD-levels. As per recent recommendations [32], total protein intake in this study based on dietary protocols averaged 1.5 g/kg/d in the EG and 1.2 g/kg/d in the CG. In a study arm not included in the present analysis, Kukuljan et al. [19] implemented a combined exercise and fortified milk (1000 mg Ca and 800 IU Vit-D/400 mL) that did not relevantly affect the exercise/bone interaction in his cohort with high baseline Ca- (911 mg/d) but low (50 IU/d) Vitamin D-intake, however. 

Unfortunately, only few studies [17,18,19,21] reported baseline dietary intake of their cohorts. Due to this limited statistical power, we do not address the effect of baseline dietary intake (e.g., Vit-D, Ca, Protein, energy intake) on the exercise bone/interaction by sub-analyses. 

#### 3.2.2. Exercise Characteristics

Table 3 specifies the exercise intervention of the included studies. Most of the RCTs compared a single exercise group with a single inactive control group. The study of Allison et al. [12] applied unilateral loading (“hopping”) and used the unloaded leg as control. Four other exercise trials implemented two exercise arms with different types of exercise interventions [15,16,22]. Another trial incorporated study arms with combined exercise and/or fortified milk supplementation [19] but was not included in the present analysis. Only one study established an active control group. This group was advised to undertake three × 30 min of unsupervised brisk walking/week [21]; however, adherence was not monitored in this CG. Although not always specified, pre-study exercise status or corresponding inclusion criteria were reported by most studies [11,12,13,14,15,17,18,19,21,22]. However, there is some evidence that men with exercise habits potentially relevant for subsequent intervention effects on BMD outcomes were included in some studies (e.g., [11,21]).

Net exercise frequency (reported exercise frequency adjusted for attendance) varied from ≈1.5 session [15,16] to ≥6 sessions/week [12]; however, most trials (Table 3) averaged between 2 and 2.5 sessions/week at least when considering attendance rate. 

### 3.3. Results on BMD at the Lumbar Spine (LS)-ROI

Nine studies with 11 exercise groups evaluated the effect of exercise versus control on LS-BMD (Figure 2). In summary, we observed a small–moderate exercise effect on LS in men (SMD: 0.38, 95% CI: 0.14–0.61) (…translated into changes in LS-BMD: 0.012 ± 0.011 g/cm^2^ or 1.2 ± 1.1%). Using the more “robust” IVhet model, the effect decreased but remained significant (SMD: 0.29, 95% CI: 0.04–0.53). There was a moderate level of heterogeneity in estimates of the exercise effect (I^2^ = 46%, Q = 18). Sensitivity analysis of imputation determined that even in the worst case (i.e., imputing with minimum correlation i.e., maximum SD) there is a significant effect (SMD: 0.35, 95%-CI: 0.11–0.59, *p* = 0.004). However, including only studies with complete data increased SMD considerably (0.68; 0.40–0.95). Results listed in Figure 2 based on imputation with mean correlation. 

Inspecting the funnel plot suggests evidence for a publication bias (Figure 3). The analysis indicates missing studies on the lower right-hand side (i.e., small studies with negative outcomes). A trim and fill analysis resulted in non-significant effects sizes (SMD: 0.12, 95%-CI: −0.16 to 0.40) after adjusting for missing studies. Further, both tests for funnel plot asymmetry, the regression (*p* = 0.02) and rank correlation test (*p* = 0.09) indicate relevant asymmetry. Applying the LFK Index (2.35), the result of significant asymmetry was confirmed. 

### 3.4. Results on BMD at the Proximal Femur

Ten studies with 13 exercise groups determined the effect of exercise on femoral neck (Figure 4) and eight studies [11,12,13,15,18,19,21,22] with 10 exercise arms addressed the total hip-BMD. Due to the higher statistical power, we decided to focus our analysis on the femoral neck ROI. In summary, we observed a small effect size for FN-BMD (SMD: 0.25, 95% CI: 0.00–0.49, *p* = 0.048) (…or 0.009 ± 0.016 g/cm^2^ (1.0 ± 1.9%)). Using the more “robust” IVhet model, the effect was slightly lower (SMD: 0.21, 95% CI: −0.04–0.47). In parallel to LS-BMD, we observed a moderate level of heterogeneity between the trials (*I*^2^ = 46%, Q= 21). Sensitivity analysis of imputation shows that when imputing with minimum correlation (i.e., maximum SD) differences between exercise and controls were non-significant (SMD: 0.16; 95% CI: −0.01 to 0.34, *p* =0.06). The same was true when including only studies with complete data (0.49; −0.53 to 1.50). Results listed in Figure 4 based on imputation with mean correlation.

In contrast to LS, we observed no evidence for a small study bias for BMD at the FN (Figure 5). Trim and fill analysis did not impute missing studies on the left side; neither regression (*p* = 0.061) nor rank correlation test indicate evidence for funnel plot asymmetry. Applying the LFK index, the result (0.78) also indicates no relevant asymmetry. 

### 3.5. Subgroup Analysis

Our subgroup analyses focused on study length (classified in 6– < 10 vs. 10–16 vs. >16 months; Table 1) and type of exercise (RT vs. WB vs. RT–WB, Table 3). In summary, no significant differences for exercise effects on LS or FN for study length (<10 months: SMD: 0.48, 95% CI: 0.12–0.85 vs. 10–16 months: SMD; 0.28, 95%-CI: −0.11 to 0.67 vs. >16 months: SMD: 0.41, 95% CI-0.19 to 1.00) and FN (0.35, −0.20 to 0.89 vs. 0.27, −0.18 to 0.72 vs. 0.09, −0.16 to 0.49) or type of exercise were observed. In detail, however, RT (LS-BMD 0.55, 0.06–1.04; FN-BMD: 0.35, −0.15 to 0.86) or combined RT–WB (LS-BMD: 0.34, 0.01–0.66; FN-BMD: 0.40, −0.11 to 0.91) protocols tended to be more favorable for improving BMD compared to WB-protocols (LS-BMD: 0.20, −0.31 to 0.72; FN-BMD: 0.00, −0.31 to 0.31). Applying the more robust IVhet model for the subgroup analysis did not lead to different results.

## 4. Discussion

In the present systematic review and meta-analysis, we provided further evidence for a favorable effect of exercise on BMD in predominately middle-aged to older men. In contrast to two recent studies [6,8] that reported positive effects for the proximal femur ROI only (Hamilton et al. [8]: SMD: 0.21, 95% CI: 0.03–0.040 for femoral neck BMD; Ashe et al. [6]: MD: 0.03 g/cm^2^, 95%-CI 0.01–0.05 g/cm^2^ for the total hip ROI.), we determined a low but significant positive effect of exercise on BMD at the FN and LS-ROI. Despite statistical differences between the studies, the main reason for the diverging results was the varying eligibility criteria. This concerns in particular pre-study health and exercise status and also pharmaceutic therapy with impact on bone metabolism. Hamilton et al. [8] rigorously excluded men with every type and volume of regular exercise prior to the intervention and, in parallel to Ashe et al. [6], did not focus on pharmaceutic therapy. In the present study, on the other hand, two reviewers carefully checked conditions (i.e., health and pre-study exercise status) and diseases/pharmaceutic therapy with relevant impact on BMD and, even more important on their potential influence on exercise effects on BMD. Although debatable, the latter resulted in the inclusion of HIV-infected men with stable antiretroviral therapy [14] or quiescent/mildly-active Crohn’s disease [11] (Table 2). However, our results indicate (Figure 2 and Figure 4) that, in diametral contrast to our expectation, positive effect on exercise induced-BMD-changes were reported for both cohorts [11,14] which relate in particular to the HIV study of Ghayomzadeh et al. [14]. In this context, we conducted two separate a priori (sensitivity) analyses (not given in the result section) without the studies of Jones et al. [11] or Ghayomzadeh et al. [14]. While the exclusion of the study of Jones et al. did not relevantly affect our results, the exclusion of the HIV-infected men with stable antiretroviral therapy [14] led to a pronounced decrease in the effect size for BMD-LS and a shift to a non-significant effect for BMD-FN.

We also included studies with men who reported pre-study exercise habits that should not or not relevantly impact the subsequent intervention effect on BMD (Table 3) [11,15,18]. Nevertheless, some studies do not report pre-study exercise status (Table 3) or pharmacologic therapy (Table 2). Consequently, there is some evidence that failure of an intervention effect on BMD might be related to (a) diseases/pharmaceutic therapy [33,34,35]), (e.g., Androgen Deprivation Therapy that induces adverse metabolic effects, including reduced muscle mass, increased fat mass and loss of bone mineral density (BMD) [36]) with striking impact on bone metabolism or/and (b) low difference between pre-intervention and intervention exercise with a corresponding lack of effective stimuli for bone. This might also relate to studies with an active control group (e.g., [21,37]) and potentially effective exercise characteristics that dilute differences between EG and CG.

Another aspect that decreases the effect sizes of exercise on any given outcome, but in particular on BMD with its complex mechanisms of action, is the “try and error approach” in phase III exercise studies [38]—an incompatible procedure in pharmacological research. Indeed, the approach of testing the effect of an intervention without (a) properly respecting basic principles of exercise application on bone ([39], e.g., [40,41]) or/and (b) determining the isolated effect of selected exercise characteristics (e.g., type of exercise, [4,42,43]) will provide at least suboptimal study results. An example for the latter aspect might be the included FrOST study [18] that focuses solely on the isolated effect of machine-based DRT on BMD in older men, whilst fully aware that the effect of a combined high impact/DRT would be potentially superior.

In addition to the traditional random-effects model approach, we applied the more robust inverse heterogeneity (IVhet) model proposed by Doi et al. [29]. The IVhet model might be the better choice to consider heterogeneity that is frequently prevalent in meta-analyses on exercise [44]. As expected, the effect sizes for LS- and FN-BMD does decrease slightly, but the result still remains significant for LS-BMD. Thus, differences in the results between the present study and the study of Hamilton et al. [8] cannot be attributed to differences in statistical procedures. We think that diverging eligibility criteria and thus the different studies included predominately account for differences in results. This may be confirmed when excluding the HIV study of Ghayomzadeh et al. [14] (see above).

Another potential limitation, but one which applies to most meta-analyses in the area of exercise and BMD, is the limited number of studies that reported (baseline) dietary parameters and/or corresponding changes of dietary habits with impact on the exercise bone interaction (e.g., Ca, Vitamin D, Protein, energy intake). This at least aggravates the decision of the author of systematic reviews and meta-analysis on excluding studies with relevant dietary effects on bone or to run sub-analyses that addresses the effect of nutritional parameters on the exercise effect on BMD at LS and FN. We recommend that future exercise trials focus on this important aspect.

Apart from generating further general evidence, another aim of the present work was to determine the relevance of selected exercise parameters on BMD effects. Considering that important exercise characteristics were either hard to categorize across the different types of exercise and with respect to given cohort (e.g., strain magnitude) or unequally distributed (e.g., BMD-status), quite homogeneous (e.g., training frequency) or simply not stated (e.g., strain rate), we used intervention length and type of exercise as moderators in our sub-analyses. In summary, we observed no significant differences between the categories; however, the statistical power for our approach was rather limited. In detail, we observed tendentially more favorable effects of RT and combined exercise protocols compared with weight bearing protocols particularly for the FN-ROI. However, three out of five WB-protocols scheduled low intensity exercise such as brisk walking or Tai Chi [17,20,22]. Nevertheless, this result confirmed the results of a recent meta-analysis with 84 included trials that did not observe relevant differences on BMD at LS and FN after resistance vs. weight bearing vs. combined exercise protocols in postmenopausal women [45]. We thus conclude that the ability of comprehensive meta-analyses to derive even raw exercise recommendations is rather limited [46,47], independently of the outcome. The close interaction between exercise parameters (e.g., strain magnitude, rate, cycle number, rest periods, training frequency [39,46]) along with the aspect that even slight differences in exercise composition might significantly modify the exercise effect on BMD [48] underscore the complexity of exercise effects on BMD that obviously collide with simple meta-analytic/meta regression approaches. A possible solution to nevertheless derive (more) dedicated exercise recommendations might be to focus on trials with comparative study arms for a given exercise parameter (e.g., exercise frequency, intensity) but otherwise identical exercise and participant characteristics (e.g., [49]).

In summary, we provided further evidence for a low, but significant effect of exercise on BMD at LS and proximal femur in men. However, we are unable to provide even rough recommendations for an exercise program dedicated to bone. Due the limited number of exercise trials with men, it might be a successful approach to determine gender differences in bone adaptation to exercise. This would legitimize the transfer of the much more extensively evaluated recommendations of exercise studies in women to men.

## Figures and Tables

**Figure 1 nutrients-13-04244-f001:**
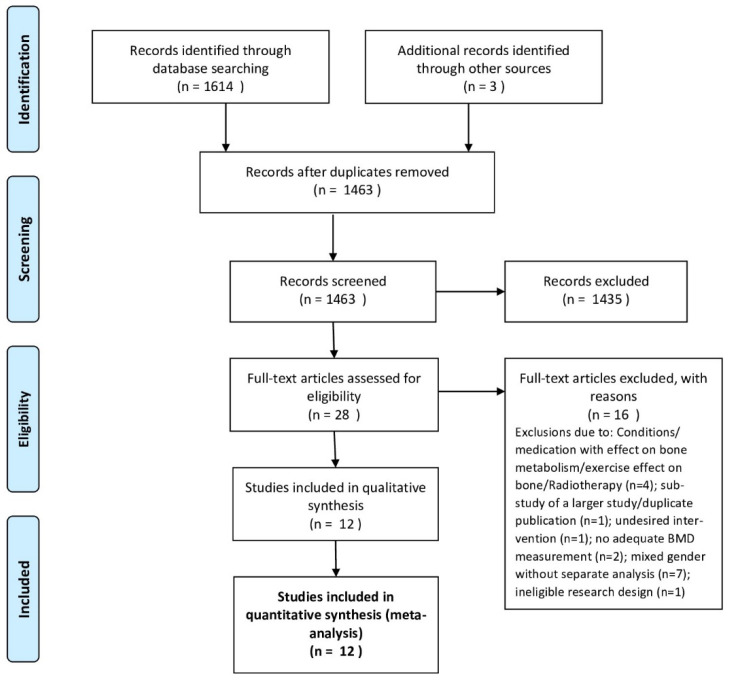
Flow diagram of search process according to PRISMA.

**Figure 2 nutrients-13-04244-f002:**
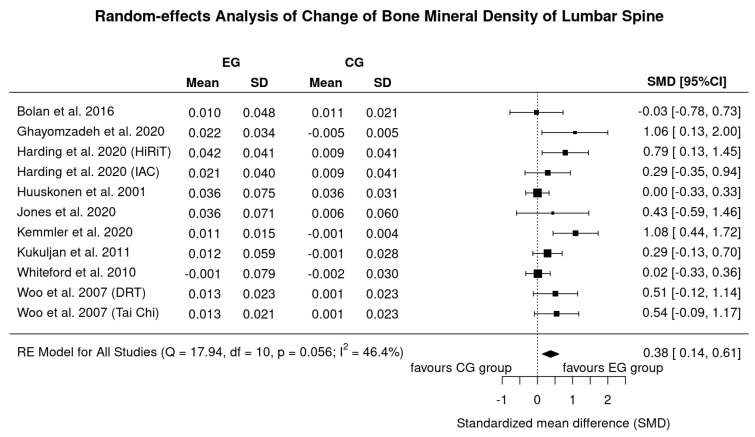
Forest plot of meta-analysis results at the lumbar spine. Data shown as pooled standard mean difference (SMD) with 95% CI for changes in exercise and control groups. CG: control group, EG: exercise group.

**Figure 3 nutrients-13-04244-f003:**
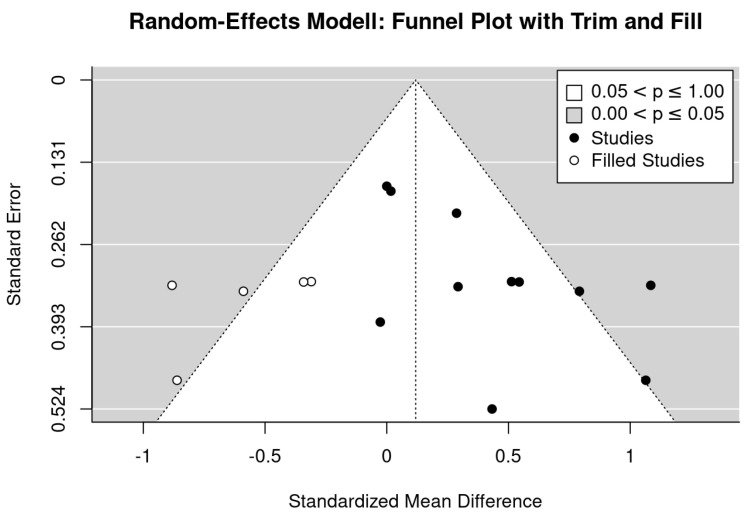
Funnel plot of the exercise studies that address lumbar spine BMD.

**Figure 4 nutrients-13-04244-f004:**
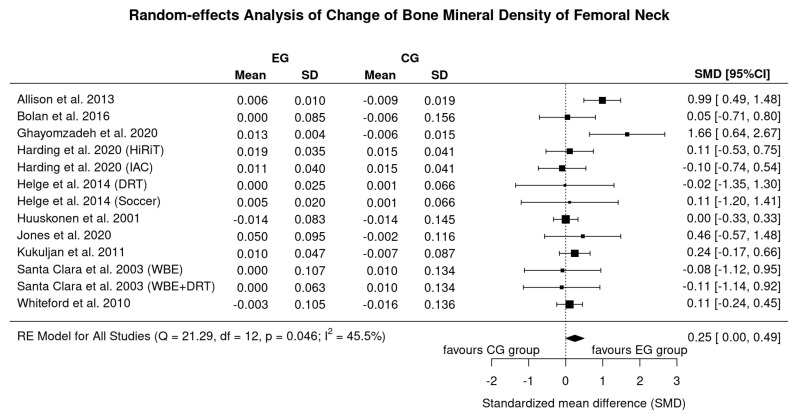
Forest plot of meta-analysis results at the femoral neck. Data shown as pooled standard mean difference (SMD) with 95% CI for changes in exercise in the control group.

**Figure 5 nutrients-13-04244-f005:**
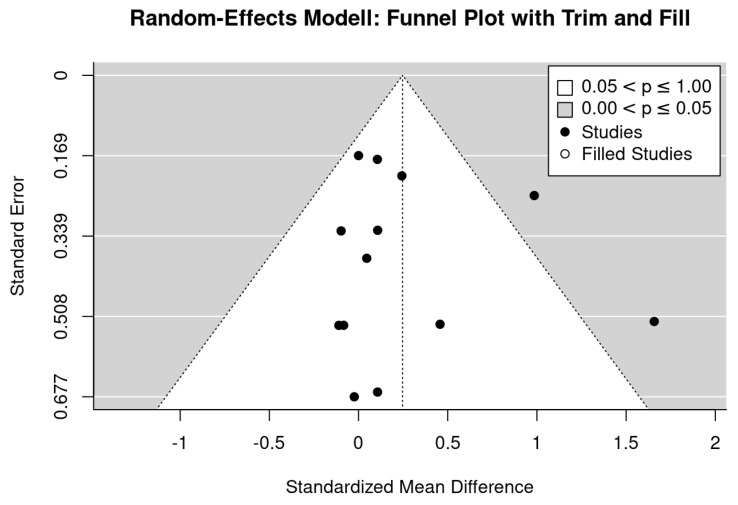
Funnel plot of the exercise studies that address femoral neck BMD.

**Table 1 nutrients-13-04244-t001:** Participant characteristics of included studies.

First Author, Year, Origin (Country)	Initial Sample Size (*n*)	Drop Out, Loss to FU (%)	Age (Years)	BMI (kg/m²)	Health Status, Bone Status	Medication with Impact on Bone
Allison, 2013 UK [12]	EG: 50	30 ^b^	69.9 + 4.0 ^b^	26.2 + 2.3 ^b^	Healthy, no BMD restriction	n.g
	CG: 50					
Bolam, 2015 Australia [13]	HI-EG: 13	23	62.1 + 6.9	25.8 + 2.8	Healthy, no osteoporosis	n.g
	CG: 14	7	58.7 + 7.4	26.6 + 3.4		
Ghayomzadeh *, 2020 Iran [14]	EG: 10	10	36.2 + 6.7	26.5 + 3.3	HIV-infected men, osteopenia or osteoporosis at hip and/or LS	No medication known to relevantly affect bone metabolism
	CG: 10	5	38.3 + 5.6	25.2 + 3.0		
Harding, 2020 Australia [15]	HiRIT-EG: 34	12	64.9 + 8.6	27.2 + 3.5	Healthy, osteopenia or osteoporosis at the hip and/or LS	Apart from 2 men in the HiRIT- and IAC-EG each, no medication known to relevantly affect bone metabolism
	IAC-EG: 33	9	69.0 + 6.8	26.6 + 4.0		
	CG: 26	19	67.4 + 6.3	26.3 + 2.8		
Helge, 2014 Denmark [16]	Soccer-EG: 9	0	68.0 + 4.0	26.1 + 3.9	Healthy, no BMD restriction	n.g
	RT-EG: 9	11	69.1 + 3.1	27.4 + 2.8		
	CG: 8	25	67.4 + 2.7	27.9 + 4.6		
Huuskonen, 2001 Finland [17]	EG: 70	6	58.1 + 2.9	27.1 ^c^	n.g, n.g. (probably healthy without BMD restrictions)	n.g
	CG: 70		58.2 + 2.9	27.2 ^c^		
Jones *, 2020 UK [11]	EG: 7	4	46.1 + 11.9	26.0 + 3.1	Quiescent or mildly-active Crohns disease, no BMD restictrion	No medication known to relevantly affect bone metabolism
	CG: 8	13	52.3 + 13.6	27.1 + 5.1		
Kemmler, 2020 Germany [18]	EG: 21	10	77.8 + 3.6	25.0 + 3.0	Sarcopenia, osteopenia or osteo-porosis at the hip and/or LS	No medication known to relevantly affect bone metabolism
	CG: 22	5	79.2 + 4.7	24.5 + 1.9		
Kukuljan, 2011 Australia [19]	EG: 46	4	60.7 + 7.1	28.1 + 3.3	n.g., partially osteopenia or osteoporosis at the hip and/or LS	No medication known to relevantly affect bone metabolism
	CG: 44	4	59.9 + 7.4	26.7 + 2.9		
Santa Clara, 2003 Portugal [20]	AE-EG: 13	n.g	57 + 11	28.1 + 4.2	Coronary artery diseases, no BMD restriction	No medication known to relevantly affect bone metabolism
	AE + RT-EG: 13		55 + 10	27.2 + 2.3		
	CG: 10		57 + 11	26.0 + 3.3		
Whiteford, 2010 Australia [21]	RT-EG: 73	11	64 + 6	26.4 + 3.1	Healthy, no osteoporosis	No medication known to relevantly affect bone metabolism
	CG: 70	4	64 + 6	26.3 + 3.0		
Woo, 2007 ^a^ Hong Kong [22]	RT-EG: 30	2	68.6 + 3.0	24.1 + 3.4	Healthy, no BMD restriction	n.g
	TaiChi-EG: 30		68.2 + 2.4	23.6 + 3.4		
	CG: 30		68.1 + 2.7	23.9 + 3.1		

*: mixed-gender Study. Values represent data from male participants; ^a^ Percentage of dropouts at 12 months, ᵇ Values represent data from exercise and control; ^c^ Calculated using body mass and height (kg/m^2^) given by the authors; AE: Aerobic exercise, CG: Control group, EG: Exercise group, HiRIT: High intensity resistance and impact training, IAC: Isometrical axial compression (machine based), LS: lumbar spine, n.g.: Not given, RT: Resistance (exercise) training, BMD: Bone Mineral Density.

**Table 2 nutrients-13-04244-t002:** Assessment of risk of bias for included studies.

	PEDro-Criteria												Additional TESTEX Criteria ¹					
Author, Year	Eligibility Criteria	Random Allocation	Allocation Concealment	Inter Group Homogeneity	Blinding Subjects	Blinding Personnel	Blinding Assessors	Participation ≥ 85%	Intention to Treat Analysis ²	Between Group Comparison	Measure of Variability	Total Score PEDro	Adverse Effects Reported	Attendance Reported	Activity Monitoring in Control Groups	Relative Exercise Intensity Constant	Exercise Volume/Energy Expended	Total Score TESTEX
Allison et al. 2013 [12]	y	+	n.a.	+	-	-	+	-	+	+	+	6	+	+	+	n.a.	+	11
Bolam et al. 2016 [13]	y	+	+	+	-	-	-	+	+	+	+	7	+	+	-	+	+	13
Ghayomzadeh et al. 2020 [14]	y	+	-	+	-	-	+	+	+	+	+	7	+	+	-	+	+	12
Harding et al. 2020 [15]	y	+	+	-	-	-	+	+	+	+	+	7	+	+	+	+	+	14
Helge et al. 2014 [22]	y	+	-	+	-	-	-	+	-	+	+	5	-	+	-	+	+	10
Huuskonen et al. 2001 [17]	y	+	-	+	-	-	-	+	-	+	+	5	-	-	-	+	+	8
Jones et al. 2020 [11]	y	+	+	-	-	-	+	+	+	+	+	7	-	+	-	+	+	13
Kemmler et al. 2020 [18]	y	+	+	-	-	-	+	+	+	+	+	7	+	+	+	+	+	14
Kukuljan et al. 2011 [19]	y	+	+	+	-	-	-	+	+	+	+	7	-	+	+	+	+	11
Santa Clara et al. 2003 [20]	y	-	-	+	-	-	-	-	-	+	+	3	-	+	-	-	+	7
Whiteford et al. 2010 [21]	y	+	-	+	-	-	-	+	+	+	+	6	+	+	-	+	+	12
Woo et al. 2007 [22]	y	+	+	+	-	-	+	+	+	+	+	8	+	+	-	-	-	10

¹ TESTEX awards one point for listing the eligibility criteria and, also in contrast to PEDro, a further point for the between group comparison of at least one secondary outcome. ² or all subjects received treatment or control as allocated. However, this aspect differs from TESTEX that specifically required an ITT analysis only.

**Table 3 nutrients-13-04244-t003:** Exercise characteristics of included studies.

Author, Year [Ref]	Exercise Status	Study Length (Months)	Progression of Intensity?	Type of Exercise, Methods	Setting/ Supervision	Intervention, Exercise Composition	Attendance	Site Specificity
Allison, 2013 [12]	Untrained	12	Yes	Unilateral “hops” Unloaded leg: CG	IE/NS	Seven session/week, five sets of ten multi-directional, unilateral hops with peak GRF of ≈3x body weight; 15 s rest between sets	91%	LS: Yes TH: Yes
Bolam, 2016 ^1^ [13]	Untrained	9	Yes	DRT (upper body) on machines and with free weights; multi-directional jumps with high GRF	JE/S IE/NS	Four sessions/week: 2 × 60 min/week: jumping (see below) and upper body DRT with four exercises. Two sets of 12 reps at 60% 1RM; two jumping sessions/week with three exercises, two–four sets, 5–18 reps and GRF: 4.6–5.8x body weight; 1 min rest between sets High volume jumping group (HV): 80 jumps/session Low volume jumping group (LV): 40 jumps/session	HVJ:53% LVJ: 65%	LS: Yes TH: Yes
Ghayomzadeh, 2020 [14]	Untrained	6	Yes	DRT (all main muscle groups) on machines and with free weights; WBE: treadmill walking/running	JE/S	Three sessions/week, eight exercises; four–twenty reps at 60–85% 1RM (i.e., first session 80–85%; second session 60–80%; third session 50–65% 1RM); each session ≈23 min of walking/running at up to ≈70%HRmax	85%	LS: Yes TH: Yes
Harding, 2020 [15]	No RT	8	Yes	DRT (deadlift, squat, and overhead press) and “jumping chin-ups”	JE/S	Two sessions/week; three exercises (deadlift, squat, and overhead press), five sets of five repetitions with 80–85% 1RM (RPE ≥ 16), five sets of five repetitions jumping chin-ups with “flat footed landing”	78%	LS: Yes TH: Yes
			Yes	Isometric-Axial-Comp-ression (IAC) at machines	JE/S	Two sessions/week, four exercises (chest press, leg press, core pull, vertical lift; bioDensity device), near-maximal 5-s isometric contraction (RPE ≥ 16)	79%	LS: Yes TH: Yes
Helge, 2014 [16]	Not given	12	Yes	Soccer (on natural grass)	JE/S	Two–three sessions/week, four set ×15 min FB at 65–90% HRmax, 2 min rest between sets.	66%	LS: Yes TH: Yes
			Yes	DRT (all main muscle groups) on machines and with free weights	JE/S	Two–three sessions/week; five–seven exercises (leg press, leg extension, leg curl, pull-down, and lateral raises, lunges, seated row) four sets at 8RM (i.e., eight reps at ≈75% 1RM), explosive concentric movement	73%	LS: Yes TH: Yes
Huuskonen, 2001 [17]	Not given	48	Yes	Brisk walking	IE/NS	Five sessions/week 60 min of brisk walking at 40–60% of VO2max (aerobic threshold pace)	Not given	LS: Yes TH: Yes
Jones, 2020 [11]	RT < 2 s/w.	6	Yes	DRT (all main muscle groups) with own body weight and elastic bands; rope skipping, multi-directional jumps	IE/ mainly NS	Three sessions/week, 5 min rope skipping, two–three sets of 10–15 reps of five different jumps (e.g., squat, broad, scissor jump); eight–ten RT exercises, two–three sets of 10–15 repetitions with “moderate-hard effort” (i.e., ≈65–75% 1RM)	62%	LS: Yes TH: yes
Kemmler, 2020 [18]	RT ≤ 45 min /w.	18	Yes	DRT (all main muscle groups) on machines	JE/S	Two sessions/week, periodized single set RT with periods of high intensity (up to 85% 1RM), high effort (by RM, supersets, drop sets) and high velocity (explosive concentric movement)	95%	LS: Yes TH: Yes
Kukuljan, 2011 [19]	Untrained	18	Yes	DRT (all main muscle groups) on machines and with free weights and jumps (IE)	JE/S IE/S	Three sessions/week, periodized RT with up to 85% 1RM and explosive velocity during the concentric phase (last 6 month), and two–three sets of different jumps with 20 reps with peak GRF of 1.5–9.7x body weight	63%	LS: Yes TH: Yes
Santa Clara, 2003 [20]	Not given	12	Yes	WBE: walking/ running on treadmill	JE/n.g.	Three sessions/week 50 min treadmill walking/running at 60–70% HRR	85%	LS: Yes TH: Yes
			Yes	DRT (all main muscle groups) on machines and treadmill walking/running	JE/n.g.	Three sessions/week 30 min treadmill walking/running at 60–70% HRR and DRT: eight exercises, two sets of eight–twelve reps at 40–50% 1RM; 2 × 20 reps of abdominal exercises; 2 × 10 reps of back exercises (intensity n.g.)	82%	LS: Yes TH:Yes
Whiteford, 2010 ^2^ [21]	≤2 s/w. ≤moderate intensity	12	Yes	DRT (all main muscle groups) on machines and with free weights	JE/S	Three sessions/week, 10 exercises, three sets at 8RM (i.e., eight reps at ≈75% 1RM)	71%	LS: Yes TH: Yes
Woo, 2007 [22]	Untrained	12	No	Tai Chi (Yang Style)	n.g.	Three session/week, 24 Forms of Yang Style, intensity n.g.	81%	LS: ? TH: yes
			No	DRT with elastic bands	n.g.	Three sessions/week, six exercises (arm lifting, hip abduction, heel raise, hip flexion, extension, ankle dorsiflexion), 30 reps with an elastic band of low–moderate strength; intensity n.g. (presumably low)	76%	LS: Yes TH: Yes

DRT: Dynamic Resistance Training; GRF: ground reaction forces; HRmax: maximum heart rate; HRR: heart rate reserve; IE: individual exercise; JE: joint (group) exercise, LS: lumbar spine; NS: non supervised; S: supervised; s/w: session/week; FN: femoral neck; WB:E weight bearing exercise; 1-RM 1-repetition maximum; ^1^ we only included results from the high volume exercise group in this analysis; ^2^ active control group (3 × 30 min of walking/week recommended).

## Data Availability

The data that support the findings of this study are available from the corresponding author (WK) upon reasonable request.

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
