# Peer review of "Exercise Effects on Bone Mineral Density in Men"

_nutrients, 2021, doi:10.3390/nu13124244_

Round 1

Reviewer 1 Report

Introduction-

The introduction needs to provide a greater rationale for why this study has been conducted. This is especially true given the two meta-analyses on this topic published in 2021 (Hamilton et al and Ashe et al), especially given that the Hamilton analysis provides very detailed information as to why an updated analysis should be conducted.  The authors discuss a compromise between eligibility and the number of participants. This seems strange given in the Hamilton analysis, which the authors suggest rigourously excluded individuals (Discussion L58), but had a similar number of participants included within the analysis (Ex: 534 FN/440 LS versus Con: 359/327) compared to the current study (461 in the intervention groups and 362 in the controls).

This study needs to provide a justification for why it started given pre-registration of the Hamilton study in 2020, with a pre-print version of the final publication available since March 2021, and this study being registered on PROSPERO in April 2021.

Methods-

The methods of analysis are ok, but they are relying on the traditional Random Effects Model that is reported throughout.  However, this model is not as robust as the IVhet model proposed by Doi et al, (Doi SA, Barendregt JJ, Khan S, Thalib L, Williams GM. Advances in the meta-analysis of heterogeneous clinical trials I: the inverse variance heterogeneity model. Contemporary clinical trials. 2015;45:130-8) and as such has been demonstrated to over-estimate summary effect statistics which may contribute to the differing results seen between the Hamilton et al analysis and this current study.  The IVhet is a more appropriate analysis to control for the hetereogeneity that is seen in exercise studies, and this has been widely discussed within the literature (e.g.  Furuya-Kanamori L, Thalib L, Barendregt JJ. Meta-analysis in evidence-based healthcare: a paradigm shift away from random effects is overdue. International Journal of Evidence-Based Healthcare. 2017;15(4):152-60).

The authors are also quite reliant on traditional methods for assessing publication bias such as Eggers regression (although this is inferred as the regression utilised is not stated).  However, the authors should consider using the Doi plot and quantitatively using the Luis Furuya-Kanamori index (LFK index). The Doi plot has been suggested to be more intuitive than the funnel plot and the LFK index more robust than the commonly used Egger’ regression-intercept test. For more information see:

  1. Furuya-Kanamori L, Barendregt JJ, Doi SA. A new improved graphical and quantitative method for detecting bias in meta-analysis. International journal of evidence-based healthcare. 2018;16(4):195-203
  2. Furuya-Kanamori L, Xu C, Lin L, Doan T, Chu H, Thalib L, et al. P value–driven methods were underpowered to detect publication bias: analysis of Cochrane review metaanalyses. Journal of clinical epidemiology. 2020;118:86-92.

Overall the methodology section is lacking detail, more information would really allow the reader to see the methodology choices made.  There are also a few points to consider:

  1. Why was the abstract/title screen only done by 1 person?
  2. Did the authors conduct any influence analysis? This would address some of the issues that the authors raise in the discussion around the “debateable” inclusion of individuals with HIV treated men and those with Crohns disease
  3. The data, search strategies, analytical code should all be made available as supplementary material for scrutiny.

Results-

These are well presented, and easy to follow which is pleasing and the forestnplots are clear./. Would be useful to label aspects from the forest plots (such as the ES [95%CI] on right hand side).

There are some strange sub-headings within the manuscript that need addressing.

Section 3.3 includes details on sensitivity analysis, which as far as I can tell there are no details within the methods section on sensitivity analysis. Was this decided a priori or post hoc. This needs to be defined by the authors.

Figure 1- would like to know the number of studies for each exclusion critiera applied to them.

Discussion-

There needs to be more critical discussion of the methods applied, and importantly why this means that the findings of increased LS BMD in this population needs to be treated with caution.  The use of the less-robust method of analysis compared to Hamilton et al, although in more participants, needs to be discussed here and put the findings of this study into context.

There also needs to be more of a summing up of what the importance of these findings are. For example, does this study change what clinicians should be utilising in this population.  Given the challenges of getting people to take part in exercise, does this study really strengthen the weight of evidence (given the methodological issues identified).

Reviewer 2 Report

The authors report important findings from a systematic review & meta-analysis of the effect of exercise on site-specific BMD. Established (PRISMA) guidelines were used in conducting the analyses. The methods and results are clearly described. Comments:

Pg 2 Line 61-62: Sentence structure needs correction.

Line 64: Is comprehensible or comprehensive the intended word?

Line 115: Tables should be numbered according to their order of citation in the text.

Figure 1: Suggest following the flow diagram structure depicted here: http://prisma-statement.org/documents/PRISMA%202009%20flow%20diagram.pdf

Table 1: please define "c" following BMI 27.1 & 27.2 for Huuskonen

1400 IE/d Vit-D - I am not familiar with this unit of measure. IU or ug is used in the USA.

Aside from the Kemmler study providing participants with calcium and vitamin D supplements and reporting protein intake, was diet controlled for in the studies? Figure 2 shows that the Kemmler study reported one of the larger beneficial effects of exercise on LS. Diet is an important factor in BMD and would assist with interpreting the findings across studies. Perhaps the authors can expand their reporting of whether dietary data were collected and included in analysis in each study. 

Round 2

Reviewer 2 Report

The authors have addressed this reviewer's comments adequately.